# Aphid facultative symbionts confer no protection against the fungal entomopathogen *Batkoa apiculata*

**Rose A. Inchauregui[1,2], Keertana Tallapragada[1,3], Benjamin J. Parker[1]***

1 Department of Microbiology, University of Tennessee Knoxville, Knoxville, TN, United States of America, 2 Department of Biological Sciences, Indiana University South Bend, South Bend, IN, United States of America, 3 UT-ORNL Graduate School of Genome Science and Technology, University of Tennessee Knoxville, Knoxville, TN, United States of America

* bjp@utk.edu

**Data Availability Statement:** All relevant data are within the paper and its Supporting Information files.

**Funding:** This work was funded by US National Science Foundation (https://www.nsf.gov) Grant

## Abstract

Fungi in the family *Entomophthoraceae* are prevalent pathogens of aphids. Facultative symbiotic bacteria harbored by aphids, including *Spiroplasma sp.* and *Regiella insecticola*, have been shown to make their hosts more resistant to infection with the fungal pathogen *Pandora neoaphidis*. How far this protection extends against other species of fungi in the family *Entomophthoraceae* is unknown. Here we isolated a strain of the fungal pathogen *Batkoa apiculata* infecting a natural population of pea aphids (*Acyrthosiphon pisum*) and confirmed its identity by sequencing the 28S rRNA gene. We then infected a panel of aphids each harboring a different species or strain of endosymbiotic bacteria to test whether aphid symbionts protect against *B. apiculata*. We found no evidence of symbiont-mediated protection against this pathogen, and our data suggest that some symbionts make aphids more susceptible to infection. This finding is relevant to our understanding of this important model of host-microbe interactions, and we discuss our results in the context of aphid-microbe ecological and evolutionary dynamics.

## Introduction

Fungal entomopathogens are important natural enemies of aphids. Field studies have found that aphids are infected by multiple species of fungi, including specialist pathogens in the family *Entomophthoraceae* (e.g., *Pandora neoaphidis*) and generalist insect pathogens like *Beauveria bassiana* [1–3]. Upon contact, fungal spores germinate and penetrate an aphid's cuticle, after which hyphal cells proliferate in the hemolymph and kill the aphid [4]. After host death, the fungus forms infective conidia that are released into the environment for transmission to new hosts. Pathogenic fungi have been studied for their potential use as biocontrol agents of plant pests like aphids [5].

The success of a fungal pathogen after aphid exposure is influenced by various factors including the host's microbiome. The microbial community of the aphid has been shown to be particularly important in resistance to the fungal pathogen *P. neoaphidis* [6]. Certain strains of

IOS-2152954 to BJP. RI was part of the NSF Research Experience for Undergraduates Program "Microbial community interactions and functions" DBI-2050743. The funders had no role in study design, data collection and analysis, decision to publish, or preparation of the manuscript.

**Competing interests:** The authors have declared that no competing interests exist.

several species of heritable (i.e. transmitted from mothers to offspring) bacteria including *Regiella insecticola*, *Rickettsia sp.*, *Rickettsiella sp.*, *and Spiroplasma sp.* make aphids less likely to become infected with *P. neoaphidis* after exposure [7]. These symbionts are referred to as 'facultative' symbionts because they are found at intermediate frequencies in natural populations and are not required for aphid survival [8]—benefiting aphids through pathogen protection is thought to be a factor in the spread and maintenance of facultative symbionts in natural aphid populations. The mechanism of this protection is not yet known, but could include the production of an anti-fungal toxin or competition for a shared resource within a host [9, 10].

Though many species of fungal pathogens infect aphids, little is known about the extent to which facultative symbionts protect against fungal entomopathogens other than *P. neoaphidis*. Parker et al. (2013) found that *R. insecticola* also protects against *Zoophthora occidentalis* [11], which like *P. neoaphidis* is a specialist pathogen (that only infect aphids) in the family *Entomophthoraceae*. That study also found no protection against the insect generalist fungal pathogen *Beauveria bassiana* and hypothesized that symbiont mediated protection might be evolving in response to the more intense selective pressure imposed by specialist rather than generalist pathogens. However, *B. bassiana* (phylum Ascomycota) is a very distantly related species to *P. neoaphidis* and *Z. occidentalis* (phylum Entomophthoromycota [12]). Here we studied the fungal pathogen *Batkoa apiculata*, which is closely related to *P. neoaphidis* and *Z. occidentalis* (which are all members of the family *Entomophthoraceae*) but is a generalist fungal pathogen that affects insect hosts across Hemiptera, Hymenoptera, and Diptera [13, 14].

We collected fungal entomopathogens from field aphids in Knoxville, TN, in 2022 and used molecular techniques to determine the species of each fungal isolate. We then established a panel of aphids that each had the same host genotype with a single infection of a facultative symbiont including strains of *Regiella* and *Spiroplasma* that we have found to be protective against *P. neoaphidis* [15–17] and *Serratia symbiotica*, which confers no protection against *P. neoaphidis*. We found that aphid facultative symbionts do not protect against *B. apiculata*, and we discuss these results within the context of aphid-symbiont-pathogen coevolution.

## Methods

### Fungal isolation

Our first objective was to isolate and collect fungal pathogens from natural aphid populations and establish these isolates in the lab for use in experiments. We collected pea aphids (*Acyrthosiphon pisum*) from Knoxville, TN, USA in 2022 from multiple host plant species, and we housed aphids from the field in Petri dishes with leaf material embedded in 2% agar. Upon aphid death and signs of fungal infection, we inverted sporulating aphids over uninfected laboratory-reared adult pea aphids (from the LSR1-01 genotype that does not harbor any facultative symbionts [18]) to expose them to spores. Exposed aphids were then housed on broad bean plants in a sealed cup cage at high humidity (>95%) for 3 days, after which they were moved to fresh plants at low humidity. Aphids that become infected with fungus produce dried resting cadavers that can be stored at 4˚C. These dried cadavers can be induced to sporulate in experiments by placing them overnight on 2% tap water agar, after which they begin to release spores.

For the fungal isolates we were able to establish in the lab, we induced sporulation from resting cadavers over a microscope slide, and visually identified fungal species by looking at primary conidia shape under a light microscope. Most isolates had the characteristic "clavate," "ovoid," or "pear-shape" of *P. neoaphidis* [4], but several isolates had spherical spores and were clearly a different species. We selected one of these isolates (collected from an aphid feeding on *Vicia sativa*) for further molecular characterization and use in experiments.

## Species identification using molecular methods

Fungi within both the genus *Batkoa* and the genus *Conidiobolus* have primary conidia that are the same size and produce similar rhizoids and are difficult to distinguish morphologically. Therefore, we used molecular methods to identify the species of fungal pathogen used in this study. We collected spores by inducing resting cadavers to sporulate as above, and extracted DNA from the spores using a Phenol-Chloroform extraction with Bender grinding buffer [19] and an ethanol precipitation. We amplified part of the ribosomal DNA sequence (the large subunit LSU) using established primers and protocols (forward: 5′-ACCCGCTGAACTT AAGC-3′) and (reverse: 5′-TCCTGAGGGAAACTTCG-3′) [20]. We excised a band at the expected size of approximately 1kb and purified the amplicon using the Zymo Gel DNA Recovery Kit, which we then sent for Sanger sequencing in both the forward and reverse directions. We generated consensus sequences by aligning the F and R sequences, and we compared the consensus sequence against the NCBI nr/nt Nucleotide collection using megaBLAST. To build a phylogeny of other species in the genus *Batkoa*, we downloaded all of the LSU sequences from NCBI in the genus *Batkoa* (or those isolates from species labeled as being from the genus *Conidiobolus* that have been subsequently re-classified into the genus *Batkoa* [21, 22]) that are from the USDA ARSEF catalog which include the type strains of each species. We built a neighbor-joining consensus tree of 10 aligned sequences (via Clustal Omega) using the HKY genetic distance model and the Bootstrap resampling method made in Geneious Tree Builder (v.2022.2.2).

## Aphid symbiont panel establishment and rearing

We used a single pea aphid (*A. pisum*) genotype in this study called LSR1-01, isolated from Alfalfa near Ithaca, NY in 1998 [18]. This line originally harbored *R. insecticola* and was cured using 1% ampicillin (which does not affect the obligate symbiont *Buchnera aphidicola* [23]). We maintained this line on fava beans in the laboratory at 20˚C with a 16L:8D photoperiod. We then established secondary symbiont infections using established protocols. Briefly, we injected a small volume of hemolymph from an infected aphid into the thorax of a 1[st] instar aphid and reared it to adulthood. The donor aphids used were aphid lines which each harbored only a single facultative symbiont infection, as determined by PCR screening using species-specific primers (S1 Table) [24]. Primers to amplify each symbiont species were adjusted to a final concentration of 0.2µM in Quick-Load *Taq* 2X Master Mix (NEB) solution. The PCR protocol had the following conditions including a 'touchdown' amplification step: An initial denaturation of 94˚C for 2 min, followed by 36 cycles of 94˚C for 20s, 56˚C declining by 1˚C per cycle until the annealing temperature is 45˚C for 50s, and 72˚C for 30s, followed by a final extension of 72˚C for 5 minutes [24].

We then collected offspring from the injected aphid late in the birth order and established lines from a single aphid. We screened the lines for the injected microbe to confirm the establishment of the infection, and waited at least 10 generations before using the lines in these experiments, at which point the lines were screened again for symbionts using PCR. Using these methods we generated an aphid panel that consisted of seven lines all with the same aphid genetic background (LSR1) but each with a single infection of a different symbiont. Four of the lines harbored *R. insecticola*: strains .313,.CF7, and.LSR were collected from pea aphids. *R. insecticola* strain .515 was collected from *Myzus persicae* [25] and has been shown in previous work to confer no protection against *P. neoaphidis* [15]. Another line harbored a fungal-protective strain of *Spiroplasma sp.* (strain .161) [16]. Finally, we included a line harboring a strain of *Serratia symbiotica*, a species that has not been shown to confer protection against fungal pathogens in aphids. We also included aphids that harbored no facultative symbionts.

Together, this panel reflects a range of phenotypic variation in terms of fungal protection against *P. neoaphidis*, including protective and non-protective strains of *Regiella*, and protective and non-protective species of other symbionts.

### Fungal infection protocol

We hydrated dried *B. apiculata* cadavers that had been stored at 4˚C by placing them on tap water agar for 12h overnight. We then infected between 139 and 154 aphids from each line in the panel across two experimental blocks, which were carried out in June and July of 2022, respectively. We subjected adult aphids (at 11 days old, as in previous studies [26]) to a spore shower from 2–3 fungal cadavers in an infection chamber, with all of the aphids from a line together in the chamber as in previous studies for other entomopathogens [27]. We rotated the fungal cadavers among the infection chambers to ensure an equal spore dose across aphid lines. We have used this protocol previously for infections with other fungal entomopathogens to control for fungal cadaver age, pathogen dose, and cadaver quality [26]. After infection, we maintained aphids on plants with cages covered in parafilm to keep the humidity high (> 95%), which is needed for entomopathogen infection. We maintained aphids at 20˚C for 48h, and then moved aphids to new plants with unvented cages. The experiment was blinded (e.g., each cage was assigned a random number so data collection was blind to treatment), and we recorded the number of fungal cadavers present in each cage at day 8 of the experiment. Though we aimed to keep the infection protocol as consistent as possible across blocks, differences in the ambient humidity in the lab and the age of the dried cadavers used for the infection (i.e. number of days stored after passaging) could have contributed to differences in infection rate between the two experimental blocks.

### Statistical analysis

We analyzed fungal sporulation using binomial logistic regression implemented in R v.4.0.2 (using the 'glm' function with family = 'binomial'). Aphid symbiont background and block were modeled as fixed effects. The overall significance of each term was determined by deriving minimal models and comparing model fit using ANOVA and Chisq statistics. Post-hoc analyses of symbiont strain/species were conducted using the Multcomp package [28], comparing infection of each aphid line with the symbiont-free line.

## Results

We obtained a 932 bp consensus sequence for our fungal isolate's LSU sequence with Sanger sequencing, which we deposited in NCBI GenBank with accession number OQ087127. Using MegaBLAST, we found a hit that was 100% identical across 100% of the query cover with "*Batkoa apiculata* strain ARSEF 3130." Like our fungal isolate, this strain is listed as being isolated from a pea aphid in the ARSEF catalog. Other BLAST hits were from fungal pathogens in the genus *Batkoa*, including *Conidiobolus pseudapiculatus* (now *B. pseudapiculatus*) [21, 22] strains ARSEF 1662 (96.46% identical) and ARSEF 395 (96.45% identical) and *C. obscurus* ARSEF 74 (from TYPE material, now called *B. obscurus* [21, 22]) at 88.29%, *B. gigantea* strain ARSEF 214 (85.65% identical) and several strains of *B. major*. A phylogenetic analysis that included these sequences placed our isolate in a well-supported clade with *B. apiculata* ARSEF 1662 (Fig 1). We should note that there was a strong blast hit (99.89% identical) to a strain labeled in NCBI as *B. obscurus* (strain CBS), but we think this record is actually a mis-identified isolate of *B. apiculata* because it is not from the ARSEF catalog, because it is significantly different from the type material for *B. obscurus*, and because there is limited detail on this

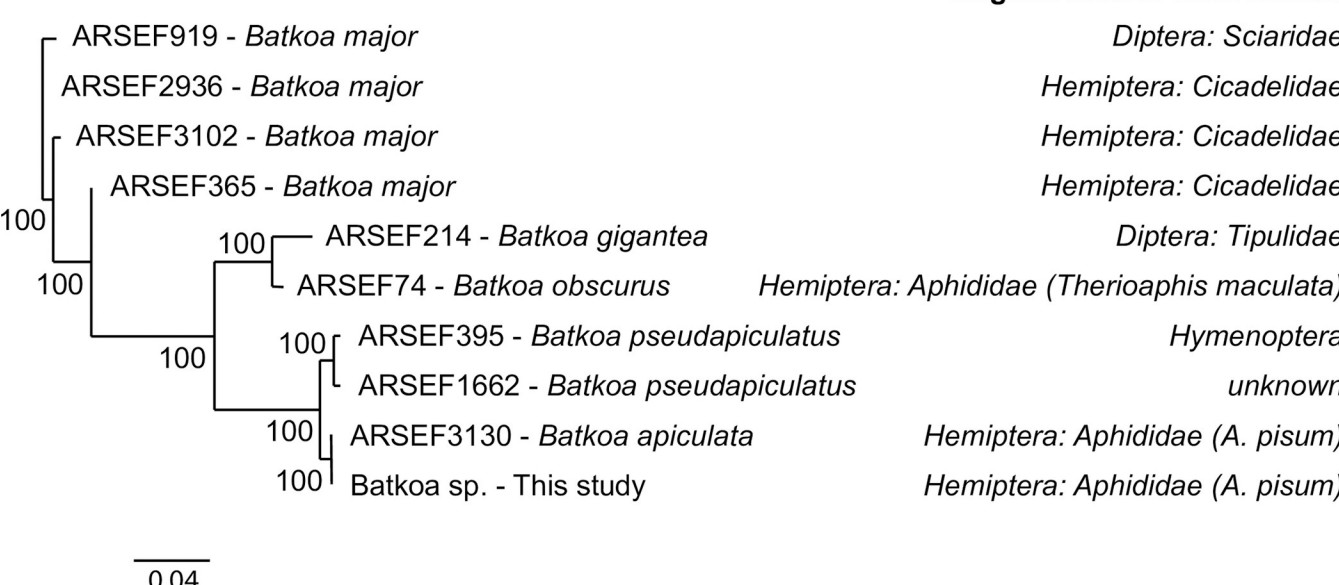

**Fig 1. Placement of the fungal isolate used in this study within the genus *Batkoa*.** A neighbor-joining consensus tree of 10 aligned sequences using the HKY genetic distance model and the Bootstrap resampling method made in Geneious Tree Builder. The phylogeny includes all of the publicly available LSU sequences of fungi in the genus *Batkoa* from the USDA ARSEF catalog. The right of the figure shows the original collection host insect species of each isolate as recorded in the catalog.

isolate available from the associated paper [22]. Based on these sequencing results and our phylogenetic analysis, we identified our fungal isolate as *B. apiculata*.

In symbiont-free aphids, approximately 20% of aphids developed a fungal cadaver by the end of the experiment (Fig 2). We found that block influenced *B. apiculata* sporulation (Chisq = 34.8, 1DF, p < 0.0001), with 31.2% and 13.6% of aphids producing a sporulating cadaver in blocks 1 and 2, respectively. Symbiont species also had a significant effect on sporulation (Fig 2; Chisq = 32.3, 6DF, p < 0.0001). Aphids harboring a 'clade 2' *Regiella* had a higher rate of sporulation over symbiont-free aphids in a post-hoc analysis (.CF7: z = 2.9, p = 0.018). Overall, our results provide no evidence of symbiont mediated protection from aphid symbionts against *B. apiculata*.

## Discussion

Interactions between aphids, facultative symbionts, and natural enemies are studied as a model for host-microbe coevolution. Though many studies have focused on symbiont mediated protection in aphids against *P. neoaphidis*, the extent to which this protection extends to other fungal entomopathogen species is less well understood. We show that aphid facultative symbionts confer no protection against the fungal pathogen *B. apiculata*, which is found in natural populations of aphids across multiple continents. In fact, we found that aphids harboring one strain of *R. insecticola* (.CF7) appear to be more susceptible to *B. apiculata* infection. This strain comes from a specific clade of *Regiella* (referred to as 'clade 2' in the literature [24]) that we have found establishes at extremely high densities within hosts [26] by suppressing aphid innate immune genes like Phenoloxidase [17, 29]. One possibility is that reduced expression of Phenoloxidase, which is an important component of aphid immunity to fungal pathogens [30, 31], makes aphids more susceptible to fungal pathogens like *B. apiculata* in the absence of symbiont mediated protection. Alternatively, the high density of these strains could impose physiological costs on hosts that make them less able to fight off fungal infection.

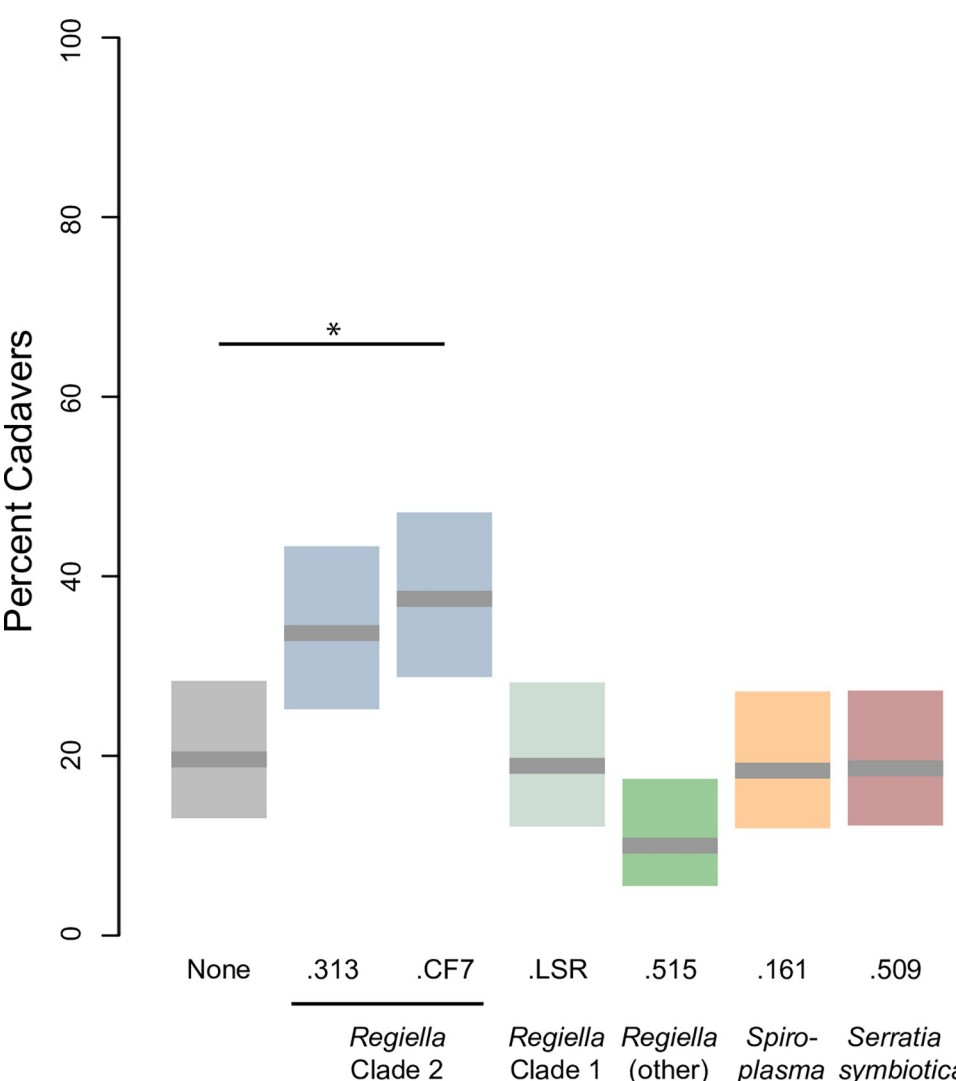

**Fig 2. Fungal infection results.** The y-axis shows the percentage of aphids that developed a characteristic *B. apiculata* cadaver. Each rectangular bar shows the 95% confidence interval, with the mean shown by the grey line. The x-axis shows the strain and species of each symbiont used in the panel. Symbiont-free aphids are shown to the left of the figure. Statistical significance at $p < 0.05$ is indicated along the top of the figure as determined by a post-hoc analysis comparing each aphid line harboring a symbiont with symbiont-free aphids.

Previous studies have found that aphid heritable symbionts confer protection against fungal pathogens in the family *Entomophthoraceae* but not against a distantly related generalist pathogen called *Beauveria bassiana* [11]. Most species in the family *Entomophthoraceae* are specialists at the level of insect family or order [32], but several species in the basal genus *Batkoa* [22], including *B. apiculata*, have been found to be generalists (and an interesting possibility raised in the literature is that generalism might be more common among basal lineages [14]). Our study therefore provides an important data point about the patterns of symbiont mediated protection against fungal pathogens in aphids: a pathogen in the family *Entomophthoraceae* that is a generalist rather than a specialist. Across several studies, there is a consistent pattern of symbiont mediated protection occurring against aphid specialist pathogens (*P. neoaphidis* and *Z. occidentalis*) but not generalists (*B. bassiana* and *B. apiculata*)—this could suggest that the more intense selective pressure imposed on hosts by specialist pathogens is needed to drive

the evolution of symbiont mediated protection. Uncovering the molecular mechanisms of fungal protection in this system would shed light on many of these questions.

## Supporting information

**S1 Table. PCR primers used for symbiont screening.**
(DOCX)

**S1 Data.**
(CSV)

## Acknowledgments

Thanks to Paula Rozo-Lopez, Matthew Kolp, Georgina Aitolo, and Mariam Sirag for technical support.

## Author Contributions

**Conceptualization:** Keertana Tallapragada, Benjamin J. Parker.

**Formal analysis:** Benjamin J. Parker.

**Funding acquisition:** Benjamin J. Parker.

**Investigation:** Rose A. Inchauregui, Keertana Tallapragada.

**Writing – original draft:** Keertana Tallapragada, Benjamin J. Parker.

**Writing – review & editing:** Rose A. Inchauregui, Keertana Tallapragada.

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
