## [Decision Letter · Decision Letter 0]

8 Feb 2023

PONE-D-23-00043Aphid facultative symbionts confer no protection against the fungal entomopathogen Batkoa apiculataPLOS ONE

Dear Dr. Parker,

Thank you for submitting your manuscript to PLOS ONE. After careful consideration, we feel that it has merit but does not fully meet PLOS ONE’s publication criteria as it currently stands. Therefore, we invite you to submit a revised version of the manuscript that addresses the points raised during the review process.

We look forward to receiving your revised manuscript.

Kind regards,

Nafiu Bala Sanda, PhD

Academic Editor

PLOS ONE

Journal Requirements:

"RI was part of the NSF Research Experience for Undergraduates Program “Microbial community interactions and functions” DBI-2050743. This work was also funded by National Science Foundation (NSF) Grant IOS-2152954 to BJP. "

"This work was funded by US National Science Foundation (https://www.nsf.gov) Grant IOS-2152954 to BJP. RI was part of the NSF Research Experience for Undergraduates Program “Microbial community interactions and functions” DBI-2050743. The funders had no role in study design, data collection and analysis, decision to publish, or preparation of the manuscript."

Reviewers' comments:

Reviewer's Responses to Questions

**Comments to the Author**

1. Is the manuscript technically sound, and do the data support the conclusions?

Reviewer #1: Yes

Reviewer #2: Partly

Reviewer #3: Yes

2. Has the statistical analysis been performed appropriately and rigorously? 

Reviewer #1: Yes

Reviewer #2: No

Reviewer #3: Yes

3. Have the authors made all data underlying the findings in their manuscript fully available?

Reviewer #1: Yes

Reviewer #2: Yes

Reviewer #3: Yes

4. Is the manuscript presented in an intelligible fashion and written in standard English?

Reviewer #1: Yes

Reviewer #2: Yes

Reviewer #3: No

5. Review Comments to the Author

Reviewer #1: It has been well known that some facultative symbiotic bacteria protect aphids from infection by certain fungal strains. However it is not clear whether this protection is universal in term of lines of aphids, strains of symbiotic bacteria and pathogenic fungi. In this manuscript, the authors present solid evidences showing such symbiont-mediated protection does not exist to infection against Batkoa apiculate. These results will lead further study on ecological and molecular mechanisms of symbiont-mediated protection in aphids.

Reviewer #2: Review of PONE-D-23-00043 - Aphid facultative symbionts confer no protection against the fungal entomopathogen Batkoa apiculata by Inchauregui, Tallapragada, and Parker

Comments for authors

I like the idea of the MS. Pea aphids have a number of microbial endosymbionts, some of which have protective functions. Here the authors focus on antifungal activities, and attempt to show that the mechanisms that protect the aphids against one common fungal pathogen may not protect against a closely related fungus. This is a limited objective, but potentially, an informative and consequential one.

I have two sets of reservations about the MS. In each case, I suspect that the authors may have the required information, but simply haven’t yet provided it in the MS. Please note, my expertise does not extend to the more molecular methods used to identify pathogens. I cannot speak to that work.

Methodology

- Pea aphids have a variety of (mostly) vertically transmitted, microbial endosymbionts including some not mentioned here (e.g., Buchnera aphidicola, Hamiltonella defensa - Buchnera is unlikely to be defensive, but is an obligate symbiont, Hamiltonella has several roles.) The bacteria mediate a number of services to the aphids, and occur in various combinations which may interact and may be functionally and mechanistically important (see refs below). This makes me ask the following methodological questions:

1) How was the aphid stock ‘cured’ of R. insecticola (lines 108-110)? - in particular, which other endosymbionts were (likely) cured in the same process, and what endosymbionts (likely) remained in the stock?

2) Did you assess the effectiveness of the endosymbiont infection injections in establishing infections with the desired endosymbionts (lines 115-118)? Similarly, did you attempt to assess which other endosymbionts (if any) were harbored by the various lines produced?

3) Were there any procedural controls that assessed the effectiveness and uniformity of success of the methods used to infect the aphids with the fungi? I ask particularly because the strongly significant block effect (line 167), when the blocks were treated as fixed variables, suggests not just that repeated blocks differed, but that block 1 differed systematically from block 2. Unless, blocks 1 & 2 differed systematically as part of the experimental design, the fungal infection rates should not have differed strongly and significantly. If that effect is real, it should be presented and explained.

Without those clarifications, it is difficult to assess the meaning of Fig. 2.

Statistics

I agree that the fungal infection/protection data should be analyzed via a generalized liner model, but:

4) We need a better description of the model used. What type of GLM was used? The methods suggest that a logistic/binomial-family GLM was most appropriate, but the symmetric error bars on Fig. 2 and the use of the ‘multcomp’ package made me suspect that a normal/gaussian GLM was used. With infection rates running as low as 10%, I suspect that the logistic GLM is much more defensible.

5) I was surprised that you chose to model the block effect as a fixed effect. Unless Block 1 really did differ systematically from Block 2 in the design of the experiment (e.g., Block 1 was done first and there was a time/duration/extinction effect), it seems more defensible to treat the blocks as random (blocking) effects in a mixed model (e.g., via the ‘glmer’ function in the lme4 package). This is particularly important when comparing the honest-to-goodness fixed effects (here the lines/symbionts) because the blocking effects, in classical theory, become part of the error variance for testing the fixed effects. The ‘glmer’ function handles this properly. - - I think we can ignore the ‘shrinkage’ effects commonly discussed in mixed model random effects because you don’t get much shrinkage with only two blocks with nearly balanced sample sizes.

6) I have always been nervous about using the ‘multcomp’ package for logistic regressions when some of the effects are close to 0 or 1 (e.g., 0.1 Fig. 2). For GLM’s, multcomp uses a likelihood ratio test that assumes the likelihood is symmetrically distributed around the MLE, which may well not be true in logistic regressions (with some extreme effects) because of the non-linear transformation of the linear result (the logit) to the probability value, and the hard bound on probabilities of infection at 0 and 1. One way to get around this is to use a Bayesian regression and directly test the ‘Bayesian p values’ comparing each line to the ‘None’ control.

Smaller points - Fig. 2

- I was confused by the ‘Percent sporulated’ label on the dependent axis. Isn’t this just percent cadavers (line 137-138)?

- The confidence intervals for each line will be asymmetric in a logistic regression, so the ‘+/- 1 SE’ error bars can be misleading. I’d suggest converting to confidence intervals.

References:

Oliver, K. M., Moran, N. A., & Hunter, M. S. (2005). Variation in resistance to parasitism in aphids is due to symbionts not host genotype. PNAS 102(36), 12795-12800.

Rock, D. I., et al.. (2018). Context-dependent vertical transmission shapes strong endosymbiont community structure in the pea aphid, Acyrthosiphon pisum. Molecular Ecology, 27(8), 2039-2056.

Smith, A. H., et al. (2015). Patterns, causes and consequences of defensive microbiome dynamics across multiple scales. Molecular Ecology, 24(5), 1135-1149.

Reviewer #3: I read the manuscript by Inchauregui et al. with interest. The paper aims to investigate the impact of the aphid facultative symbiont against the the fungal entomopathogen Batkoa apiculata. In general the topic is relevant however there is some comments:

I suggest authors to motivate the choice of facultative bacteria tested in this study. On the basis of what they were chosen? Please improve this aspect because it is the purpose of the work.

Section Material and Methods:

Line 109: Please explain how you can verify the infection status of both subclones cured and infected aphid with Regiella insecticola. The infection of the aphid was verified using diagnostic PCR? If yes, please give a full description of the method.

Line 110: please add your laboratory conditions used in this experiment. Aphids were reared on V. faba seedlings at which temperature? Photoperiod?

Line 113: you used species-specific primers? Give details please? And what about other facultative symbionts such as Hamiltonella defensa? You are sure that your aphid without this bacteria? Have you used specific primers?

Line 128: are you sure for the age of your aphids used in this experiment? 11 days old?!!! I think it’s too old.

Section Discussion:

The discussion section is very short and should be supported with previous work. For example:

1.The aphid facultative symbiont Serratia symbiotica influences the foraging behaviors and the life-history traits of the parasitoid Aphidius ervi (Entomologia Generalis)

2. Effects of pea aphid secondary endosymbionts on aphid resistance and development of the aphid parasitoid Aphidius ervi: A correlative study. (Entomologia Experimentalis et Applicata)

6. PLOS authors have the option to publish the peer review history of their article (what does this mean?). If published, this will include your full peer review and any attached files.

Reviewer #1: No

Reviewer #2: No

Reviewer #3: No

---

## [Author Response · Author response to Decision Letter 0]

7 Mar 2023

Please see the response to reviewer document

---

## [Decision Letter · Decision Letter 1]

19 Apr 2023

PONE-D-23-00043R1Aphid facultative symbionts confer no protection against the fungal entomopathogen Batkoa apiculataPLOS ONE

Dear Dr. Parker Benjamin,

Thank you for submitting your manuscript to PLOS ONE. After careful consideration, we feel that it has merit but does not fully meet PLOS ONE’s publication criteria as it currently stands. Therefore, we invite you to submit a revised version of the manuscript that addresses the points raised during the review process.

We look forward to receiving your revised manuscript.

Kind regards,

Nafiu Bala Sanda, PhD

Academic Editor

PLOS ONE

Journal Requirements:

Additional Editor Comments (if provided):

The manuscript can be accepted pending minor revision suggested by the reviewers.

Reviewers' comments:

Reviewer's Responses to Questions

**Comments to the Author**

1. If the authors have adequately addressed your comments raised in a previous round of review and you feel that this manuscript is now acceptable for publication, you may indicate that here to bypass the “Comments to the Author” section, enter your conflict of interest statement in the “Confidential to Editor” section, and submit your "Accept" recommendation.

Reviewer #1: (No Response)

Reviewer #4: (No Response)

Reviewer #5: All comments have been addressed

Reviewer #6: (No Response)

2. Is the manuscript technically sound, and do the data support the conclusions?

Reviewer #1: (No Response)

Reviewer #4: Yes

Reviewer #5: Yes

Reviewer #6: Yes

3. Has the statistical analysis been performed appropriately and rigorously? 

Reviewer #1: (No Response)

Reviewer #4: Yes

Reviewer #5: Yes

Reviewer #6: Yes

4. Have the authors made all data underlying the findings in their manuscript fully available?

Reviewer #1: (No Response)

Reviewer #4: Yes

Reviewer #5: Yes

Reviewer #6: Yes

5. Is the manuscript presented in an intelligible fashion and written in standard English?

Reviewer #1: (No Response)

Reviewer #4: Yes

Reviewer #5: Yes

Reviewer #6: Yes

6. Review Comments to the Author

Reviewer #1: (No Response)

Reviewer #4: In their paper, Inchauregui et al. studied the influence of facultative symbionts on the ability of aphids to resist a generalist entomopathogenic fungus. They showed that the facultative symbionts tested did not confer any protection against the fungus Batkoa apiculata. Although this is more of a case study (only one species of fungus tested) with negative results that are difficult to generalize, the study is very well done and addresses an area of research that remains understudied. Despite this and its concise form, the study is well written, to the point, and sets the stage for studying the mechanisms behind these protective effects. I have a few minor comments below.

L48: Perhaps also cite Hansen et al. 2012 (1).

L77: Where do these uninfected aphids exposed to spores come from? From the field? Are these aphids infected by facultative symbionts?

L113-114: Was the screening done before or after the 10 generations were obtained? It is not clear to me if it was done after (which is necessary to ensure that the relationship between host and symbiont has stabilized).

L120: Do you know more about the Serratia symbiotica 509 strain? Are there any studies that have characterized the associated induced phenotypic effects?

L128: The aphids are 11 days old. They are young adults. But is this the same developmental stage as in previous studies that showed protective effects against entomopathogenic fungi?

L186-187: The subject behind "strains" is not clear. I assume that the authors are referring to Regiella. But since only one strain of Regiella has been tested, the subject should be expressed in the singular.

I have reread the article several times and have no further comments. Perhaps the discussion could have been more in depth (2), but perhaps it is best to keep it concise here due to the brevity of the experience. Perhaps raise the idea that fungal pathogens can switch from parasitism to mutualism as in cicadas with an interest in studying these interactions in other Hemiptera (3–6). I would have many more thoughts to share, but I would stray from the basic article. I hope the authors will teach us more about insect-symbiont-fungus interactions.

1. Hansen AK, Vorburger C, Moran NA. 2012. Genomic basis of endosymbiont-conferred protection against an insect parasitoid. Genome Res 22:106–114.

2. Gibson CM, Hunter MS. 2010. Extraordinarily widespread and fantastically complex: comparative biology of endosymbiotic bacterial and fungal mutualists of insects. Ecol Lett 13:223–234.

3. Huang Z, Zhou J, Zhang Z, He H, Wei C. 2023. A Study on Symbiotic Systems of Cicadas Provides New Insights into Distribution of Microbial Symbionts and Improves Fluorescence In Situ Hybridization Technique. 3. Int J Mol Sci 24:2434.

4. Matsuura Y, Moriyama M, Łukasik P, Vanderpool D, Tanahashi M, Meng X-Y, McCutcheon JP, Fukatsu T. 2018. Recurrent symbiont recruitment from fungal parasites in cicadas. Proc Natl Acad Sci 115:E5970–E5979.

5. Michalik A, Franco DC, Deng J, Szklarzewicz T, Stroiński A, Kobiałka M, Łukasik P. 2023. Variable organization of symbiont-containing tissue across planthoppers hosting different heritable endosymbionts. Front Physiol 14.

6. Kobiałka M, Michalik A, Walczak M, Szklarzewicz T. 2018. Dual “Bacterial-Fungal” Symbiosis in Deltocephalinae Leafhoppers (Insecta, Hemiptera, Cicadomorpha: Cicadellidae). Microb Ecol 75:771–782.

Reviewer #5: (No Response)

Reviewer #6: (No Response)

7. PLOS authors have the option to publish the peer review history of their article (what does this mean?). If published, this will include your full peer review and any attached files.

Reviewer #1: No

Reviewer #4: No

Reviewer #5: No

Reviewer #6: **Yes: **Chen Luo

---

## [Author Response · Author response to Decision Letter 1]

19 Apr 2023

Please see the response to reviewers document for our responses.

---

## [Editor Report · Decision Letter 2]

9 May 2023

Aphid facultative symbionts confer no protection against the fungal entomopathogen Batkoa apiculata

PONE-D-23-00043R2

Dear Dr. Benjamin Parker,

We’re pleased to inform you that your manuscript has been judged scientifically suitable for publication and will be formally accepted for publication once it meets all outstanding technical requirements.

Kind regards,

Nafiu Bala Sanda, PhD

Academic Editor

PLOS ONE

Additional Editor Comments (optional):

Having addressed the concerned raised by the reviewers, the manuscript can be accepted for publication in PLOS ONE, congratulations!
---

## [Editor Report · Acceptance letter]

11 May 2023

PONE-D-23-00043R2 

Aphid facultative symbionts confer no protection against the fungal entomopathogen *Batkoa apiculata*

Dear Dr. Parker:

I'm pleased to inform you that your manuscript has been deemed suitable for publication in PLOS ONE. Congratulations! Your manuscript is now with our production department. 

Kind regards, 

on behalf of

Dr. Nafiu Bala Sanda 

Academic Editor

PLOS ONE